# Skin-Grafting and Dendritic Cell “Boosted” Humanized Mouse Models Allow the Pre-Clinical Evaluation of Therapeutic Cancer Vaccines

**DOI:** 10.3390/cells12162094

**Published:** 2023-08-18

**Authors:** Bijun Zeng, Davide Moi, Lynn Tolley, Natalie Molotkov, Ian Hector Frazer, Christopher Perry, Riccardo Dolcetti, Roberta Mazzieri, Jazmina L. G. Cruz

**Affiliations:** 1Peter MacCallum Cancer Centre, Melbourne, VIC 3000, Australia; 2Frazer Institute, The University of Queensland, Brisbane, QLD 4102, Australia; 3Faculty of Medicine, The University of Queensland, Brisbane, QLD 4072, Australia; 4Department of Otolaryngology, Princess Alexandra Hospital, Brisbane, QLD 4102, Australia; 5Sir Peter MacCallum Department of Oncology, The University of Melbourne, Melbourne, VIC 3010, Australia; 6Department of Microbiology and Immunology, The University of Melbourne, Melbourne, VIC 3010, Australia

**Keywords:** cancer, vaccine, T cells, animal models, dendritic cells, humanized-mouse models, HPV, melanoma, breast cancer

## Abstract

Vaccines have been hailed as one of the most remarkable medical advancements in human history, and their potential for treating cancer by generating or expanding anti-tumor T cells has garnered significant interest in recent years. However, the limited efficacy of therapeutic cancer vaccines in clinical trials can be partially attributed to the inadequacy of current preclinical mouse models in recapitulating the complexities of the human immune system. In this study, we developed two innovative humanized mouse models to assess the immunogenicity and therapeutic effectiveness of vaccines targeting human papillomavirus (HPV16) antigens and delivering tumor antigens to human CD141^+^ dendritic cells (DCs). Both models were based on the transference of human peripheral blood mononuclear cells (PBMCs) into immunocompromised HLA-A*02-NSG mice (NSG-A2), where the use of fresh PBMCs boosted the engraftment of human cells up to 80%. The dynamics of immune cells in the PBMC-hu-NSG-A2 mice demonstrated that T cells constituted the vast majority of engrafted cells, which progressively expanded over time and retained their responsiveness to ex vivo stimulation. Using the PBMC-hu-NSG-A2 system, we generated a hyperplastic skin graft model expressing the HPV16-E7 oncogene. Remarkably, human cells populated the skin grafts, and upon vaccination with a DNA vaccine encoding an HPV16-E6/E7 protein, rapid rejection targeted to the E7-expressing skin was detected, underscoring the capacity of the model to mount a vaccine-specific response. To overcome the decline in DC numbers observed over time in PBMC-hu-NSG-A2 animals, we augmented the abundance of CD141^+^ DCs, the specific targets of our tailored nanoemulsions (TNEs), by transferring additional autologous PBMCs pre-treated *in vitro* with the growth factor Flt3-L. The Flt3-L treatment bolstered CD141^+^ DC numbers, leading to potent antigen-specific CD4^+^ and CD8^+^ T cell responses *in vivo*, which caused the regression of pre-established triple-negative breast cancer and melanoma tumors following CD141^+^ DC-targeting TNE vaccination. Notably, using HLA-A*02-matching PBMCs for humanizing NSG-A2 mice resulted in a delayed onset of graft-versus-host disease and enhanced the efficacy of the TNE vaccination compared with the parental NSG strain. In conclusion, we successfully established two humanized mouse models that exhibited strong antigen-specific responses and demonstrated tumor regression following vaccination. These models serve as valuable platforms for assessing the efficacy of therapeutic cancer vaccines targeting HPV16-dysplastic skin and diverse tumor antigens specifically delivered to CD141^+^ DCs.

## 1. Introduction

Vaccines are one of the most successful medical advancements in human history, with remarkable examples of the broadly applied vaccines against the hepatitis B virus, human papillomavirus (HPV), and SARS-CoV-2 [1]. Over time, the concept of vaccination has evolved to include approaches that deliver disease-specific antigens to not only prevent but also treat overt diseases like cancer. Therapeutic cancer vaccines aim to generate ex novo or expand pre-existing anti-tumor immunity by targeting antigen-specific T cells and skewing their responses to a more favorable cytotoxic phenotype [2,3]. However, in comparison with prophylactic vaccines, this treatment modality has shown, at best, modest efficacy in the clinic. Better preclinical mouse models capable of recapitulating the hallmarks of the disease and predicting responses in humans are fundamental to overcoming the limited immunogenicity and potency of current cancer vaccines, thereby improving their clinical applicability and efficacy. Among these models, humanized mice engrafted with human immune cells and bearing human or human-like tumors have been a valuable tool [4,5]. To this end, we generated two humanized-mouse models to test therapeutic vaccines targeting epithelial lesions expressing HPV16 antigens and to characterize the immunogenicity and efficacy of vaccines delivering any antigen directly to specialized-dendritic cell (DC) subsets *in vivo* representing a step up from their murine counterparts.

One of the most robust animal models to study HPV-driven neoplastic lesions is the murine skin-grafting model based on the transgenic K14E7 mice, which expresses cancer–associated tumor HPV16 oncoprotein E7 under the control of the keratinocyte-restricted K14 promoter [6]. In this model, transgenic K14E7 skin is transplanted onto the flank of immunocompetent syngeneic non-transgenic mice (*C57BL/6*). Importantly, although the E7 protein is immunogenic in both animals and humans [7,8], K14E7 skin grafts are not rejected by the host’s immune system, thus recapitulating the immunosuppression seen in human HPV-associated neoplasia [6,9]. The main advantage of this model versus subcutaneous injection of HPV-oncogene-expressing cell lines is that the expression of E7 by keratinocytes leads to epithelial hyperplasia, immune cell infiltration, and an overall transcriptome profile comparable to that seen in patients with HPV-associated cervical intraepithelial neoplasia stage 3 (CIN3) [10]. Therefore, the skin grafting K14E7 system is an ideal model to test therapeutic HPV vaccines as it recapitulates hallmarks of the human disease, including the need for successful vaccines to overcome the immunoregulatory effects of the HPV16 E7 expression. 

However, despite promising results in mouse studies, vaccine candidates showing efficacy against HPV16 E7-expressing lesions have failed to elicit the same curative responses in clinical trials, highlighting the incongruences between mouse and human biological systems, especially at the immune system level [11,12]. These issues underscore the limitation of mouse models and the need for complementary systems that more faithfully mimic human immune response toward HPV16-derived lesions and HPV16 vaccines. Therefore, we generated a new humanized skin graft model where HLA-A*02^+^ tumors were exposed to HPV16 E7 antigens and human immune cells that could then mount an anti-tumor response upon vaccination with a therapeutic DNA HPV vaccine.

Despite continuous advances, the choice of the best antigen formulation for human cancer vaccines remains a challenge, and suitable preclinical models are required to investigate the immunogenicity of new human tumor antigens and functionally characterize the CD4^+^ and CD8^+^ T cell responses elicited *in vivo*. Moreover, the recent development of nanoparticle-based delivery systems selectively targeting distinct DC subpopulations also requires the availability of humanized mouse models suitable to investigate the *in vivo* DC targeting and efficacy of novel vaccination platforms. This is the case of vaccines targeting human CD141^+^ DCs through nanoparticles functionalized with moieties that bind Clec9A. These DCs are particularly effective in presenting tumor antigens through MHC class I (cross-presentation) and MHC class II molecules, thus eliciting strong CD8^+^ and CD4^+^ antigen-specific T cell responses [13,14]. Therefore, we generated a second mouse model that allows the assessment of the efficacy of therapeutic vaccines delivering different human tumor antigens to human Clec9A^+^ DCs. 

## 2. Materials and Methods

### 2.1. Mice

C57BL/6J (H-2b) mice were obtained from the Animal Resources Centre (ARC, Perth, WA, Australia) and C57BL/6J mice transgenic for the HPV16 E7 oncoprotein driven from a K14 promoter (designated as K14E7 mice) from an inbred colony at the Translational Research Institute (Brisbane, Australia). NSG (NSG; strain NOD.Cg-PrkdcscidIl2rgtm1Wjl/SzJl; Stock #005557; RRID:IMSR_JAX:005557) and NSG-A2 mice carrying the human HLA Class I-A2 variant (NOD.Cg-Mcph1Tg(HLA-A2.1)1Enge Prkdcscid Il2rgtm1Wjl/SzJ;Stock#009617; RRID:IMSR_JAX:009617) were purchased from The Jackson Laboratory (Bar Harbor, ME, USA). All animal work was performed in accordance with the regulatory standards of the National Health and Medical Research Council (NHMRC) and the Australian Code for the care and use of animals for scientific purposes. All animal protocols were pre-approved by the University of Queensland Animal Ethics Committee (approval number: UQDI/198/18, AE—2022/AE000029 and UQDI/252/16).

### 2.2. PBMC Isolation and Cryopreservation

Buffy Coats from healthy donors were received from the Australian Red Cross within the frames of an approved HREC and a signed contract of collaboration. Buffy coats were diluted (1:4) in PBS (GIBCO by ThermoFisher, Scoresby, VIC, Australia), and the leukocyte fraction was enriched using standard Ficoll-Paque PLUS (Cytiva by Merk, Bayswater, VIC, Australia) gradient centrifugation. The PBMC interface was then harvested and washed twice with PBS. The expression of HLA-A*02 was confirmed by flow cytometric analysis. Afterward, PBMCs were immediately injected into non-irradiated NSG-A2 mice for engraftment or cryo-preserved. PBMCs were resuspended and cryopreserved in freezing media, (10% dimethylsulfoxide (DMSO; Sigma-Aldrich by Merk, Bayswater, VIC, Australia) and 90% filtered and inactivated fetal bovine serum (FBS; GIBCO by ThermoFisher, Scoresby, VIC, Australia)). Cryotubes, containing 5 million cells, were subjected to a controlled freezing process using Corning^®^ CoolCell^®^ Freezing Containers (Corning, Silverwater, NSW, Australia). The containers were placed at −80 °C overnight, followed by long-term storage in liquid nitrogen. 

### 2.3. The Humanized K14E7 Skin Grafting Model

The NSG-A2 recipient mice were double-grafted as previously described [9,15]. Briefly, whole ears from lethally irradiated (1000 cGy) donor mice (NSG-A2_C57BL/6J and NSG-A2_K14E7) were surgically removed and split into dorsal and ventral surfaces. The remaining cartilage tissue was gently removed with a scalpel, and two dorsal sheets were placed onto an ~1cm^2^ incision on each thoracic flank of fully anesthetized non-irradiated NSG-A2 recipients. The grafts were covered with antibiotic-permeated Vaseline gauze (Bactigras, Smith and Nephew, London, UK) and covered with micropore tape and Flex-wrap (Lyppard, QLD, Australia). The bandages were removed 7–8 days after surgery, and the grafts were monitored for the duration of the experiment. Photographs, including a ruler, were taken three times a week after the grafts were well healed (~1 month after surgery). The graft’s area was analyzed using FIJI v2.9 Imaging software. For the humanization of the animal, the grafted NSG-A2 recipients were intravenously injected with 5 × 10^6^ HLA-A*02 fresh PBMCs.

### 2.4. TNE Preparation

Clec9A-targeting tailored nanoemulsions (TNEs) loaded with a pool of neoepitopes were prepared as previously described [14]. Briefly, peptides were dissolved in ultrapure water to a final protein concentration of 1 mg/mL and emulsified with Span 80 (1%, *w*/*v*) (Sigma-Aldrich by Merk, Bayswater, VIC, Australia). The resulting emulsion (1 mL) was frozen rapidly in dry ice for 2 h before being lyophilized for 24 h. The resulting pellet was dissolved in Miglyol 812 (20 µL) (Cremer Oleo, Hamburg, Germany) and used as an oil phase for preparing Clec9A-targeting tailored nanoemulsions (TNEs) loaded with peptides as described in [14]. In some experiments, TNEs were labelled using the Dil fluorescent dye (Invitrogen^TM^, Scoresby, Australia) as previously described [14].

### 2.5. Dendritic Cell-Boosted Humanized Models 

NSG-A2 mice were humanized (PBMC-hu-NSG-A2) via i.v. injection of 5 × 10^6^ freshly isolated HLA-A*02 PBMCs on the indicated days. Cryopreserved PBMCs from autologous HLA-A*02 healthy donors were thawed in RPMI-10 (GIBCO by ThermoFisher, Scoresby, VIC, Australia) with DNase I (0.1 mg/mL) (Roche by Merk, Bayswater, VIC, Australia) and resuspended in RPMI-10 (2 × 10^7^ cells/mL). The PBMCs were incubated with recombinant human Flt3-L (200 ng/mL) (Peprotech^®^, Scoresby, VIC, Australia) at 37 °C, 5% CO_2_ for 24 h. The cells were then washed and resuspended in PBS, and 2 or 5 million Flt3-L-treated PBMCs in 200 μL PBS were i.v. injected into the humanized mice the day before TNE injection.

### 2.6. Humanized Tumor Models

NSG-A2 mice were injected with either 5 × 10^6^ MDA-MB-231 cells orthotopically in the shaved mammary fat pad or 5 × 10^6^ A375 cells subcutaneously in the shaved right flank. Six days after tumor cell injection, mice were i.v. injected with 5 × 10^6^ freshly isolated HLA-A*02 PBMCs. The day before the first and second vaccinations, the mice were i.v. injected with 2 million autologous Flt3-L-treated PBMCs generated as described above. The mice were vaccinated on the indicated days via i.v. injection of 100 µL Clec9A-targeting TNEs loaded with the pools of epitopes listed in Table 1.

### 2.7. Flow Cytometry

To immune profile PBMCs after being transferred into the mice, blood and spleens were collected at different time points. The spleens were finely chopped, and a single-cell suspension was obtained by passing the sample through a 70 µm nylon strainer (Corning^®^). Erythrocytes from blood and spleens were eliminated with ACK lysis buffer (0.15 M NH_4_Cl, 1 mM KHCO_3_, 0.1 mM EDTA; Sigma-Aldrich by Merk, Bayswater, VIC, Australia). The samples were then washed and resuspended in PBS. Prior to antibody staining, the mouse and human FC receptors were blocked using mouse and human TruStain FCX (BioLegend^®^, San Diego, CA, USA) at 1:300 dilution in PBS for 15 min on ice. The Life/Dead fixable Aqua Dead cell stain kit was used according to the manufacturer’s recommendations (Invitrogen^TM^ by ThermoFisher, Scoresby, VIC, Australia). The staining of the surface markers was performed using the combinations of fluorescently labelled monoclonal antibodies listed in Table 2 and diluted 1:200 in FACS buffer (1× PBS, 2% fetal bovine serum, 0.05 mM EDTA) and incubated for 1 h on ice in the dark. For intracellular staining, the samples were fixed and permeabilized using eBioscience™ Foxp3/Transcription Factor Staining Buffer Set following the manufacturer’s instructions (eBioscience by ThermoFisher, Scoresby, VIC, Australia). The intracellular staining of cytokine was performed using fluorescently labelled mAbs (α-human-TNFα #cat502916; α-human-IFNγ #cat506526; α-human-IL2 #cat500320; BioLegend, San Diego, CA, USA) diluted 1:100 in FACS buffer and incubated for 1 h at room temperature. Before the acquisition, the samples were washed three times with cold FACS buffer and resuspended in 200 µL of FACS buffer. Flow cytometry was performed using a BD LSR Fortessa^TM^ X20 (BD Biosciences^®^, Mulgrave, VIC, Australia), and the results were analyzed using the FlowJo^TM^ 10.9.0 software (BD Biosciences^®^, Mulgrave, VIC, Australia).

### 2.8. Immunohistochemistry and Immunofluorescence

The presence of infiltrating human hemopoietic CD45^+^ cells (monoclonal rabbit anti-human CD45 antibody, 1:500, #ab40763; Abcam, Cambridge, UK) and T cells (XP^®^ monoclonal rabbit anti-human CD3ε (D7A6E™), 1:300, #85061; Cell Signalling Technology^®^ Danvers, MA, USA) in murine tissue was detected by immunohistochemistry (secondary Goat-anti-Rabbit IgG H&L (HPR) antibody, 1:1000, #ab 205718; Abcam, Cambridge, UK) and immunofluorescence (secondary Goat-anti-Rabbit IgG H&L (Alexa Fluor^®^) antibody, 1:6500, #111-585-003; Jackson Immunoresearch, West Grove, PA, USA) in Formalin-Fixed Paraffin-Embedded tissue blocks as previously described [16,17].

### 2.9. Polyfunctional Assay

Spleen samples were harvested, minced into small pieces (~0.2 cm^2^) using a scalpel blade, and cultured in 5 mL digestion buffer (1 mg/mL collagenase D, 20 μg/mL of DNase I; Roche by Merk, Bayswater, VIC, Australia) in RPMI 1640 with 10% FBS (RPMI-10 media) for 1 h at 37 °C. The enzymatic digestion was terminated by EDTA (1 mM final concentration). The digested spleens were then passed through a 70 μm cell strainer into a 50 mL falcon tube and washed with 10 mL RPMI-10. Erythrocytes were lysed by adding 1 mL of ACK lysis buffer for 1 min at RT. Splenocytes were resuspended in RPMI-10 at a final concentration of 2 × 10^7^ cells/mL. 

To set up the T cell poly-functional assay, 100 µL splenocytes (2 × 10^6^ cells) were plated per well into 96-well round bottom plates and cultured with 10 μg/mL of the indicated peptide mix (Table 1 or Table 3). Cells treated with PMA (100 ng/mL) (Caymen Chemical, Ann Arbor, MI, USA) and ionomycin (1 µg/mL) (Sigma-Aldrich by Merk, Bayswater, Australia) or medium served as positive and negative controls. Brefeldin A (5 µg/mL; BioLegend, San Diego, CA, USA) was added to each well 1 h after the start of the culture, and the cells were incubated for 16 h at 37 °C. A Live/Dead fixable Aqua Dead cell stain kit was used according to the manufacturer’s recommendations (Invitrogen^TM^). The cells were then stained for mouse CD45, human CD45, CD3, CD4, and CD8 cell surface markers (Table 4), permeabilized, and fixed using an eBioscience^TM^ Foxp3/Transcription Factor Fixation/Permeabilization kit following the manufacturer’s protocol (eBioscience, by ThermoFisher, Scoresby, VIC, Australia). Thereafter, the cells were stained for IFN-γ, TNF-α, and IL-2 (Table 4). The production of one, any combination of two, or three cytokines among the CD4^+^ and CD8^+^ T cells was quantified by flow cytometry to determine the presence and poly-functionality of tumor-specific T cells (Table 1) or HPV vaccine-specific T cells (Table 3). Co-expression of multiple cytokines was analyzed by Boolean Combination Gates (FlowJo^TM^ 10.9.0 software (BD Biosciences^®^, Mulgrave, VIC, Australia). The gating strategy is depicted in Appendix A.

### 2.10. Ex Vivo Re-Stimulation of Engrafted PBMCs

The spleens from PBMC-hu-NSG-A2 animals 21 days post-engraftment were processed as described in the section “Flow cytometry”. Single-cell splenocytes were re-stimulated *ex vivo* and incubated for 16 h at 37 °C with Dynabeads™ Human T-Activator CD3/CD28 (Gibco^TM^ by by ThermoFisher, Scoresby, VIC, Australia) following manufacture’s protocol and in the presence of Brefeldin A (5 µg/mL)). Thereafter, the cells were stained for IFN-γ, CD4, CD8, and CD3 markers (Table 3) as per the “Polyfunctional assay” section.

### 2.11. Statistical Analysis

For the FACS and vaccination experiments, the statistical analysis was conducted using GraphPad Prism software (v9; Boston, MA, USA). An unpaired two-tailed *t*-test with equal SD was performed to analyze samples when only two groups were compared. The significance between more than two experimental groups was evaluated by conducting an ordinary one- or two-way analysis of variance (ANOVA) test with equal SD. Bars indicate the mean, and error bars depict the SEM. * *p* < 0.05, ** *p* < 0.01, *** *p* < 0.001, and **** *p* < 0.0001.

## 3. Results

### 3.1. Engraftment and Immune Profile of Transferred PBMCs into NSG-A2 Mice

One of the main conditions that limit the use of humanized mouse models is the often-suboptimal engraftment of human immune cells in the host mouse. Therefore, we sought to evaluate whether the use of fresh or frozen HLA-A*02-matching PBMCs had an impact on the percentage of human hematopoietic cells (hCD45^+^) with respect to the total CD45^+^ cells (murine + human CD45^+^ cells) in the blood of engrafted NSG-A21 (PBMC-hu-NSG-A2) mice two weeks after cell transfer (Figure 1A). The use of freshly isolated PBMCs significantly improved the percentage of blood hCD45^+^ from ~20% in the frozen PBMC group up to 80% (Figure 1B). Using fresh PBMCs, a >70% engraftment was maintained in the blood of recipient animals up to 28 days post-transference. Interestingly, in the spleen, the percentage of hCD45^+^ cells increased approximately fourfold over time from 20% in week 1 to ~80% in week 4, suggesting progressive colonization of the peripheral organs from blood circulating PBMCs and/or the passive expansion of certain populations in the organ.

Using lineage-specific markers, we characterized the main cell subsets of the innate and adaptive immune responses in the blood and spleens of grafted animals at 7, 14, and 28 days post-cell transference (Figure 1C). Myeloid cells (CD11b^+^CD3ε^−^CD56^−^CD19^−^), double-negative T cells (DN; CD3ε^+^CD56^−^CD4^−^CD8α^−^), Natural Killer cells (NK; CD56^+^CD3ε^−^CD19^−^), and NKT cells (CD56^+^CD3ε^+^CD19^−^) followed a similar trend of decreasing abundance in the blood and spleen over time (Figure 1C). 

Conversely, B cell (CD19^+^CD3ε^−^CD56^−^) ratios diminished in blood but increased in the spleen over time from ~6% to ~18% of the total engrafted hCD45^+^ cells. The T cell compartment accounted for the vast majority (>50% on day 8) of all engrafted human cells, both in blood and in the spleen. The CD4^+^ T cell lineage (*red*) increased in the second week in the blood to remain stable in both blood and tissues. In contrast, the CD8^+^ T cell lineage (*blue*) reached significantly higher levels than those found in the PBMC donor’s blood (dotted line), ratios that remained constant across the analyzed organs. Interestingly, the abundance of DP^+^ T cells (CD4^+^CD8^+^CD3ε^+^CD56^−^) increased over time in the blood and spleens of engrafted animals, being previously correlated with the onset of graft-versus-host disease (*GvHD*) [18].

We then sought to characterize the engrafted immune subpopulations using unbiased clustering based on the lineage markers (Figure 1D). Twenty-one clusters were identified across the main immune cell subsets, two of which were classified as “Others”, as the selected markers did not provide an annotation. The main CD4^+^ T cell population in the donor PBMCs expressed high levels of CD4 coreceptor and CD3ε chain (cluster 1). A second population with intermediate CD3ε levels was present in donor PBMCs and engrafted mice, especially in the spleen at late time points (cluster 2). Two CD11b^+/−^ CD4^+^ T cell populations with intermediate levels of CD11c were exclusively present in the engrafted animals (clusters 3 and 4). The different CD8^+^ T cell phenotypic repertoires were more widely represented across donor PBMCs and murine samples, with three shared populations of T cells being CD8^high^, CD8^low^, and CD8^high^CD11c^low^ (clusters 5, 8, and 6), the latter in accordance with a more active phenotype [19,20,21,22,23,24]. Similar to what was observed for the CD4^+^ compartment, a highly activated CD8^+^CD11c^low^CD11b^high^ population (cluster 7) was only present in engrafted human cells in the blood at day 28 and in the spleen at all analyzed time points. Interestingly, a distinct CD8^+^ T cell subset expressing low levels of CD11c and the B cell marker CD19 (cluster 9) was only observed in the spleen 4 weeks post-transference. This typically B cell marker (CD19) has been reported to be expressed by peripheral T cell lymphomas, which, although not linked to neoplasia in our model, suggests the activation of non-conventional pathways [25]. The last two T cell clusters were identified as DP T cells (CD3ε^+^CD4^+^CD8α^+^) (cluster 10) and DN (CD3ε^+^CD8α^−^CD4^−^) (cluster 11). By day 28 post-cell transference, both NK (cluster 12) and NKT cells, including CD8α positive and negative subsets (clusters 13 and 14), disappeared from the engrafted animals. The expression of different levels of CD19 identified four B cell clusters (clusters 15–18), one of them exclusive of the humanized spleens (cluster 18). This subset was characterized by the high expression of CD11c. Only one cluster of myeloid cells defined by the high expression of the CD11b^+^ pan-marker was identified (cluster 19).

As T cells are the main targets of vaccines, we last investigated whether this population was capable of being activated three weeks post-engraftment. To do so, splenocytes from PBMC-engrafted NSG-A2 mice were isolated and ex vivo stimulated with anti-CD3ε/anti-CD28 antibody-coated beads (Figure 1E). As a readout of the intensity of the response to the stimulation, the intracellular accumulation of IFNγ was measured by flow cytometry. Both CD4^+^ and CD8^+^ T cell subsets expressed high levels of the effector cytokine, demonstrating their capacity to respond to T cell receptor-mediated stimulation up to 21 days post-transference.

### 3.2. Dysplastic Skin-Grafting Humanized Mouse Model and Response to DNA-HPV Therapeutic Vaccine

To engineer an HLA-A*02 humanized mouse model for therapeutic HPV vaccine testing, we first induced the expression of the HPV16 E7 oncoprotein in NSG-A2 mice by crossing K14E7 transgenic, and non-transgenic mice as a control, with NSG-A2 animals (Figure 2A). The resultant F1 (NSG-A2_K14E7) expressed both HLA-A*02 molecules and HPV16 E7, which led to the characteristic thickening of the skin epidermis, as previously described in the parental K14E7 strain [6,9]. To avoid the transference of any cells from the skin donor, F1 NSG-A2_B6/K14E7 donors were lethally irradiated before the ear skin was transplanted onto the flank of NSG-A2 recipient mice, which received 5 × 10^6^ (5 M) allogeneic HLA-A*02 PBMCs once the grafts were well healed (minimum 1-month post-surgery) (Figure 2B). 

The grafts showed no signs of rejection, as measured by graft area, before or after the PBMC transference (Figure 2C). Additionally, the model was still viable up to two months post-PBMC reconstitution with no weight loss, which is the first indication of *GvHD* onset (Figure 2D). HPV-expressing E7 grafts collected from transplanted animals 2 months after surgery displayed the expected hyperplastic epithelium (Figure 2E) significantly thicker than control grafts (Figure 2F). Importantly, human cells were able to colonize the skin grafts of both F1 NSG-A2_B6 and _K14E7 donors as well as the spleen (Figure 2G) where the majority of infiltrating cells were T cells (Figure 2H), the main population responsible for graft rejection in previous murine models [6]. 

Last, we tested the capacity of the model to respond to therapeutic HPV vaccination and to direct an anti-tumor response toward HPV-expressing lesions measured by the reduction of the graft area (Figure 2I). To this end, we employed the experimental DNA vaccine AMV002, which encodes an E6–E7 fusion protein from high-risk HPV16. Previously, we showed that immunization prior to the grafting of T cell receptor transgenic E7TCR269 mice, with enhanced numbers of E7-specific T cells, promoted humoral and cellular immunity, which led to significant K14E7 graft rejection in this murine model [26]. Additionally, this vaccine has been well tolerated and elicited a humoral response in a phase I dose escalation study (ACTRN12618000140257) [27]. Hence, two days after the transplantation of 2 M fresh-HLA-A*02^+^ allogeneic PBMCs (*n* = 12), skin-grafted animals were vaccinated intradermally in the pinna of the ear with 30 ng of HPV DNA vaccine (AMV002) or the DNA vector expressing an irrelevant protein (NTC-gD2) once a week for 3 consecutive weeks. In contrast to NTC-gD2 vaccinated animals or NSG-A2_B6 grafts, only NSG-A2_K14E7 grafts rapidly responded to the first and second HPV vaccinations with a significant reduction of the initial area of the graft (Figure 2I). Furthermore, the expression of IFNγ upon ex vivo re-stimulation with E7 peptides was observed exclusively in CD4^+^ and CD8^+^ human T cells isolated from the spleens of animals vaccinated with the HPV vaccine (Figure 2J). 

Overall, our new skin-grafting humanized mouse model recapitulated the expected determinants of the HPV-driven disease, including epithelial hyperplasia, and displayed a responsive human immune system to anti-tumor vaccination capable of mounting an antigen-specific response demonstrated by the sole shrinkage of the E7-expressing lesion. 

### 3.3. Boosting PBMC-hu-NSG-A2 Mice with Flt3-L-Treated PBMCs Increases the Number of Human CD141^+^ cDC1 Cells and Enhances Antigen-Specific T Cell Responses 

We next investigated whether a humanized and HLA-matched model with good engraftment of functional CD4^+^ and CD8^+^ T cells could be used to characterize the immunogenicity and test the therapeutic efficacy of novel cancer vaccine platforms targeting human tumors. 

Our previous work indicated that the functionalization of TNEs with F-actin, the natural ligand of human Clec9A, allows the targeting and activation of human CD141^+^ cDC1 cells expressing Clec9A [14]. Humanized mouse tumor models suitable to investigate the feasibility, immunogenicity, and therapeutic efficacy of vaccines targeting human Clec9A^+^ DCs would require the presence of adequate numbers of this critical population of human DCs. Engraftment of NGS mice with unmanipulated donor PBMCs is not expected to provide such numbers, considering the very low percentage (~0.03%) of circulating CD141^+^ DCs [28] (Figure 1C). To overcome this limitation, we explored the possibility to increase the number of CD141^+^ DCs by boosting PBMC-hu-NSG-A2 mice with autologous PBMCs pre-treated *in vitro* with human Fms-like tyrosine kinase receptor 3 ligand (Flt3-L), known to expand CD141^+^ DCs both *in vitro* and *in vivo* [29,30]. Both 24 h and 48 h *in vitro* incubations of human PBMCs with 200 ng/mL of human Flt3-L increased the proportions of CD141^+^ DCs (Figure 3A). However, because cell viability dropped significantly after 24 h, 24 h treatment was chosen for subsequent experiments.

We then investigated whether F-actin-TNEs can efficiently target and activate human CD141^+^ DCs in boosted PBMC-hu-NSG-A2 mice boosted with autologous FLT3-L-treated PBMCs. To this end, the NSG-A2 mice were humanized with 5 × 10^6^ HLA-A*02^+^ PBMCs on day 0 and boosted with FLT3-L-treated autologous PBMCs on days 7 and 14 (Figure 3B). The mice were then vaccinated on days 9 and 16 with F-actin-TNEs loaded with the highly immunogenic pan-HLA-DR NY-ESO-1_119–143_ [31] and the HLA-A*02 NY-ESO-1_157–165_ epitope peptides [32]. The F-actin-TNEs were labelled with the Dil fluorescent dye for *in vivo* tracking. The control mice were vaccinated with empty F-actin-TNEs. The analysis carried out at the end point (day 17) demonstrated that a detectable fraction of human CD141^+^ DCs were able to uptake F-actin-TNEs (Figure 3B) and be specifically activated by NY-ESO-1 epitopes, as shown by the significant CD86 up-regulation (Figure 3C).

We also investigated the ability of F-actin-TNEs loaded with the NY-ESO-1 epitopes to elicit specific T cell responses. The spleens from humanized mice were collected 5 days after the second vaccination (d21) and total splenocytes containing the human CD45^+^ lymphocytes were re-stimulated ex vivo with the NY-ESO-1 epitopes. The vaccination resulted in the generation of strong epitope-specific CD8^+^ and CD4^+^ T cell responses, as shown by the expression of IFNγ, IL-2, or TNFa (Figure 3D and Appendix A). Importantly, a significantly higher number of epitope-specific CD8^+^ T cells expressing one or two cytokines was observed in PBMC-hu-NSG-A2 mice receiving two boosts of autologous FLT3-L-treated PBMCs in comparison with mice not receiving these boosts (Figure 3D). These findings are consistent with stronger, and polyfunctional antigen-specific human T cell responses elicited in “boosted” PBMC-hu-NSG-A2 mice. The same immunization scheme efficiently generated CD8^+^ T cell responses specific for different tumor-associated HLA-A*02 epitopes including Survivin_95–104_, Mammaglobin_83–92_, HER3_356–364_, and cMET_654–662_ (Figure 3E) [33,34,35].

Repeated injections of high numbers of human PBMCs could accelerate the onset of *GvHD*, an invariable occurrence in PBMC-hu-NSG mice that may limit the use of these models for long-term investigations. To limit and/or delay the onset of *GvHD* and, therefore, allow for an adequate time window to perform vaccination experiments, we investigated whether boosting humanized mice with fewer PBMCs could reduce the risk of *GvHD* development. We compared PBMC-hu-NSG-A2 mice boosted with two i.v. injections (at days 9 and 14 after engraftment) of 2 × 10^6^ (2 M) or 5 × 10^6^ (5 M) autologous Flt3-L-treated HLA-A*02^+^ PBMCs. 

No significant weight loss was observed with either 2 M or 5 M PBMCs for up to 40 days, highlighting a delayed *GvHD* onset in this HLA-matched model (Figure 3F). Notably, the efficiency of human CD45^+^ cell engraftment was significantly higher in PBMC-hu-NSG-A2 mice receiving two boosts with 2 × 10^6^ autologous PBMCs (Figure 3F). This condition was therefore adopted for all the subsequent experiments.

### 3.4. Direct Comparison between PBMC-hu-NSG and PBMC-hu-NSG-A2 Models Supports the Use of HLA Matching Systems to Test Cancer Vaccines

Humanization of NSG-A2 with HLA-A*02-matching PBMCs has been previously used to generate strong immune responses and to delay *GvHD* onset [5]. To obtain direct evidence supporting these statements in vaccines targeting Clec9A^+^ DCs, we compared NSG and NSG-A2 strains side by side for the efficiency of vaccine-induced T cell responses (Figure 4A). To this end, groups of HLA.A*02^+^ PBMC-hu-NSG-A2 or -NSG mice (*n* = 8) were vaccinated with two weekly i.v. injections of F-actin-TNEs loaded with universal tumor-associated HLA-A*02 epitopes (hTERT_540–548_, Survivin_95–104_) [33,36] and the HLA Class II promiscuous Survivin_97–111_ [37]. Seven days after the second vaccination (24 days after PBMC engraftment), the spleens were harvested, and polyfunctional T cell responses were assessed in human CD45^+^ cells after ex vivo peptide re-stimulation. Although the global extent of epitope-specific T cell responses induced in the two mouse strains was similar, a slightly but significantly higher number of monofunctional CD8^+^ T cell responses was observed in the PBMC-hu-NSG-A2 mice (Figure 4A and Appendix A). 

Next, we investigated whether the time of onset of *GvHD* and the therapeutic efficacy of TNE vaccination differed between tumor-bearing “boosted” PBMC-hu-NSG and -NSG-A2 models. The NSG and NSG-A2 mice bearing orthotopically injected MDA-MB-231 triple-negative breast cancer cells were humanized with HLA-A*02^+^ PBMC 6 days after tumor cell injection (d0) and boosted with two injections of Flt3-L-treated autologous PBMCs on days 9 and 16 (Figure 4B). The mice were then vaccinated with F-actin TNEs loaded with the hTERT_540–548_, Survivin_95–104_, and Survivin_97–111_ epitopes on days 10, 17, and 24 [33,36,37]. As shown in Figure 4B (right panel), both non-vaccinated (*control*) and vaccinated (*TNE*) MDA-MB-231-bearing PBMC-hu-NSG mice showed signs of *GvHD* starting on day 24 after PBMC injection. Conversely, all tumor-bearing PBMC-hu-NSG-A2 mice showed no evidence of *GvHD* up to 6 weeks after PBMC engraftment. Importantly, despite similar tumor growth kinetics in the two mouse strains, the efficacy induced by the vaccination in PBMC-hu-NSG could not be reliably assessed due to an earlier onset of *GvHD*. Conversely, F-actin-TNE vaccination resulted in a significant MDA-MB-231 tumor growth inhibition in the PBMC-hu-NSG-A2 mice at 30 days in which no sign of *GvHD* occurred (Figure 4B, left panel). 

Last, we tested the suitability of our optimized boosted PBMC-hu-NSGA2 mice model to assess the immunogenicity and therapeutic efficacy of an experimental cancer vaccine in an additional tumor model: the A375 melanoma model. Hence, A375 melanoma-bearing humanized NSG-A2 mice received two weekly boosting and three weekly vaccinations with Clec9A-targeting F-actin-TNE loaded with a pool of tumor-associated immunogenic epitopes, including the HLA-A*02 Survivin_95–104_, Cyclin I_71–80_ (RNA edited, Zhang, 2018), MAGE-A3_271–279_, Tyrosinase_369–377_, STAT1_VLW_, and the promiscuous HLA Class II Tyrosinase_386–406_ and NY-ESO-1_119–143_ epitopes (Figure 4C). The vaccinated mice showed significant inhibition of tumor growth (Figure 4C), which was associated with strong CD4^+^ and CD8^+^ T cell responses specific to the tumor antigens targeted by the vaccine (Figure 4D and Appendix A). No sign of *GvHD* was observed up to day 40, and similar levels of engraftment of human CD45^+^ cells were retained in both untreated and vaccinated mice, although slightly reduced as compared with what was observed on day 17 (Appendix A). Overall, these results further confirmed the suitability of the boosted PBMC-hu-NSGA2 mice for the investigation of the immunogenicity and therapeutic efficacy of cancer vaccines targeting cDC-1 cells to elicit a strong anti-tumor response. 

## 4. Discussion

The concept of humanized mouse models dates back to the 1980s when the discovery of Prkdc^scid^ (severe combined immunodeficiency (SCID)) mutation paved the way for xenotransplantation of human tissues in mice [38]. Since then, humanized mouse models have been widely used in biological and medical research to study human diseases and test therapeutic approaches [39]. For immune assays, PBMCs or hematopoietic stem cells (HSCs) from umbilical cord blood, bone marrow, fetal liver, or adult mobilized HSCs can be used to reconstitute the immune system of humanized mice. Among these, PBMCs are the most widespread reconstitution method due to their high numbers and easy accessibility and manipulation. Here we have shown that the engraftment of fresh PBMCs was superior to their thawed counterparts. Our findings indicate that the cryopreservation process affects cell viability and the capacity of PBMCs to engraft the host. In support, the use of temperatures lower than −150 °C and long storage times have a significant impact on the expression profile of thawed PBMCs triggering the up-regulation of “stress genes”, which can significantly modify the cells’ phenotype [40]. Cryopreservation induces alterations in the cellular composition of PBMCs following thawing, particularly affecting innate immune cells such as monocytes and NKs, as well as diminishing the proliferative capacity of T cells [41]. Therefore, based on our results, it is recommended to use fresh PBMCs when a high engraftment ratio is required for downstream experiments. Researchers should be aware of the impact that cryopreservation has on transplanted cells and consider employing appropriate sample preparation and storage methods when fresh cells are not an option.

Another aspect to take into consideration is the dynamic changes in the frequency and phenotype of the transplanted cells occurring over time. We observed that as previously reported, the dominant engrafted population was T cells, both in the blood and the spleen, and that new subsets within this lineage, including double-positive T cells, related to the onset of *GvHD* [18] and CD11c^+^ T cells progressively expanded over time. The expression of the α integrin CD11c on T cells has been linked to a highly activated T cell phenotype, which positively correlated in our animals with CD11b up-regulation, a sign of recent cell activation [24]. This model was also capable of sustaining high levels of B cells in the spleen, which have been previously reported to produce IgGs detectable in the blood of engrafted animals [42]. Notably, one of the B cell subsets present in the engrafted animals showed high expression of CD11c^+^. In both humans and mice, CD11c^+^ B cells accumulate in autoimmune diseases, such as rheumatoid arthritis, after malaria infection with age and show a memory phenotype capable of quickly differentiating into antibody-secreting cells [23,43,44,45]. Overall, T and B cells showed signs of activation upon engraftment, in some cases within the first week after transplantation (i.e., CD11c^+^ CD11b^+/−^ CD4^+^ T cells). Conversely, other innate cell populations, including myeloid cells and NKs, rapidly became undetectable from the blood and the spleen. This fact has led to the development of other models where the human cytokines and growth factors are supplemented exogenously or directly by transgenic mouse strains expressing stem cell factor (SCF), colony stimulator factor-1 (CSF-1), granulocyte monocyte-CSF (GM-CSF), interleukine-3 (IL-3), IL-6, IL-7, and IL-15 to enhance the survival of distinct myeloid and lymphoid cell populations [5,46].

The high prevalence of T cells in PBMC-hu-mice makes it an ideal model to test the efficacy of vaccine formulations against tumors sensitive to effector CD8^+^ T cells. This is the case for the K14E7 skin-grafting murine model, where the transference of E7-specific CD8^+^ T cells and vaccination induced the rejection of E7-expressing skin grafts [6]. In this study, a humanized version of the murine model was developed, where skin dysplasia was induced by the expression of the HPV-16 E7 oncogene in keratinocytes carrying the HLA-A*02 allele. Like the immunocompetent murine model, keratinocytes act as the source of tumor antigen, but whether they are responsible for direct presentation is under debate. Recent evidence suggests that under inflammatory conditions, human keratinocytes can present antigens to naive CD8^+^ T cells skewing the response to Th1/Th17 phenotypes [47]. In support, keratinocytes can up-regulate the expression of MHC-I and class II molecules to present antigens not only to CD8^+^ but also to CD4^+^ T cells in response to pro-inflammatory stimuli [48]. Therefore, we speculate that E7-expressing keratinocytes in the skin graft could act as non-professional presenting cells and activate human CD8^+^ T cells via HLA-A*02 molecules. However, myeloid cells from transferred PBMCs may contribute to T cell activation by engulfing dead cells from the grafted skin. Either way, our experiments demonstrated that human immune cells could infiltrate the skin grafts in our HLA-A*02 model and that only the E7-expressing epidermis showed a significant reduction upon the DNA-HPV vaccine, indicating that this model can be used to validate HPV-therapeutic vaccines in a humanized immune system context.

The presence of professional human antigen-presenting cells in humanized mouse models is an absolute prerequisite to reliably investigate the immunogenicity and efficacy of novel cancer vaccines designed to deliver tumor antigens to distinct human DC subpopulations *in vivo*. However, the development of suitable humanized models must consider the low numbers of functionally relevant DCs, such as human CD141^+^ cross-presenting DCs, normally present in the fresh donor-derived PBMC preparations used for engraftment.

The results presented herein demonstrate that this limitation can be overcome by boosting PBMC-hu-NSG-A2 mice with autologous PBMC pre-treated *in vitro* with Flt3-L, which is critical to expanding cDC1 populations and their progenitors [49]. Although present in small numbers, this subset of DCs is critical to promote anti-tumor immune responses, and its exploitation together with stimulatory agents, immune checkpoint inhibitors, or CAR T cells was associated with improved therapeutic outcomes [49,50]. Our boosting approach resulted in the detection of activated human CD141^+^ DCs taking up F-actin-TNEs in the spleen of PBMC-hu-NSG-A2 mice as well as in a higher number of antigen-specific CD8^+^ T cells. Globally, the results obtained demonstrate that the boosted PBMC-hu-NSG-A2 mouse model is suitable to target Clec9A DCs, which are emerging as major players in the onset of anti-tumor responses [51], and to investigate the extent and quality of CD4^+^ and CD8^+^ T cell responses specific for a variety of different tumor epitopes. 

The transmembrane receptor Cle9A (C-type lectin domain family 9 member A) binds F-actin exposed by necrotic cells, thus enabling the capture of dead-cell-associated antigens for presentation to CD8^+^ and CD4^+^ T cells. The correlation between low Clec9A expression and poor outcomes in lung adenocarcinoma and breast cancer highlights the important role played by cells expressing this receptor. Clec9A internalizes upon ligand binding and induces phagosomal membrane rupture, allowing antigen access to both MHC I and II presentation [51]. Our boosted humanized model allows the testing of novel therapies targeting this important subset of DCs.

The onset of *GvHD* is one of the major limitations in the use of mice humanized by the adoptive transfer of human PBMC [52]. The mature and functional human T cells react against the mouse tissue causing damage and organ failure manifested as rapid body weight loss and death. The relatively rapid development of *GvHD* in standard NSG mice significantly reduces the time window available for investigating any therapeutic effect mediated by immunotherapies that are known to require time to be effective. Here we show that by creating an HLA-matched system where transgenic NSG mice expressing the human HLA-A2 allele (NSG-A2 mice) are humanized with PBMC derived from HLA-A*02 donors and injected with human HLA-A*02 tumors, we significantly delay *GvHD* onset. This provides sufficient time to test the efficacy of immunotherapies such as cancer vaccination including those directly targeting Clec9A^+^ DCs *in vivo*. The model also allows reliable measurement of the extent of vaccine-induced tumor growth inhibition together with the quantitative and qualitative monitoring of tumor-specific CD4^+^ and CD8^+^ T cell responses. Our model opens the possibility to investigate combination therapies including vaccines and immune checkpoint inhibitors, which are known to synergize with antigen-delivering approaches. 

## 5. Conclusions

In conclusion, by exploring the impact of HLA matching in the NSG system, the PBMC source used for engraftment, and the DC complementation strategies, we have developed two novel humanized mouse models suitable to characterize the immunogenicity and therapeutic efficacy of vaccines targeting HPV antigens expressed by dysplastic skin or delivering diverse tumor antigens to human Clec9A^+^ DCs *in vivo*. The models can be applied to a variety of cancer types and vaccine modalities relevant to immuno-oncology and allow the preclinical assessment of cancer vaccines necessary to support their clinical application. 

## Figures and Tables

**Figure 1 cells-12-02094-f001:**
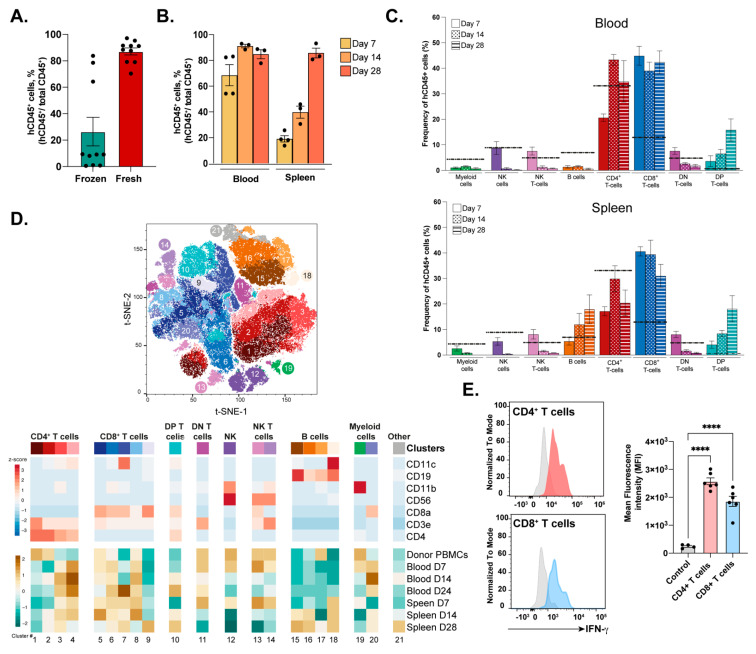
**Engraftment and immune profile of transferred PBMCs into NSG-A2 mice by flow cytometry.** (**A**) Proportion of human PBMCs (hCD45^+^) from total human and mouse hematopoietic cells (total CD45^+^) in the blood of engrafted NSG-A2 animals 2 weeks post-transference. (**B**) Percentage of engrafted hCD45^+^ cells in blood and spleen at 7, 14, and 28 days post-transference. (**C**) Frequency of several immune cell subsets derived from engrafted human PBMCs at 7, 14, and 28 days post-transference in the blood and spleen of the recipient mice. The dashed line represents the frequency of a particular cell subset in the donor PBMCs. (**D**) Unbiased clustering based on the expression of seven surface markers represented by different colors on the t-distributed stochastic neighbor embedding plot (t-SNE-1, *top*) and heat map of the relative fluorescence intensity of a given marker (*bottom*). The bottom heat map represents the relative abundance of a given cell subset within the hematopoietic compartment and across different time points (D7, D14, and D28) in the donor PBMCs, blood, and spleen samples. Cluster IDs are indicated by numbers. (**E**) Histogram representation (*left*) and quantification (*right*) of IFNγ intracellular expression by *ex vivo* unstimulated T cells (grey) or stimulated CD4^+^ and CD8^+^ T cells extracted from the spleen at 21 days post-PBMC transference. NK, natural killers; DN, double-positive T cells. The data are presented as mean values ± SEM. *p* < 0.0001 (****) (One-way ANOVA).

**Figure 2 cells-12-02094-f002:**
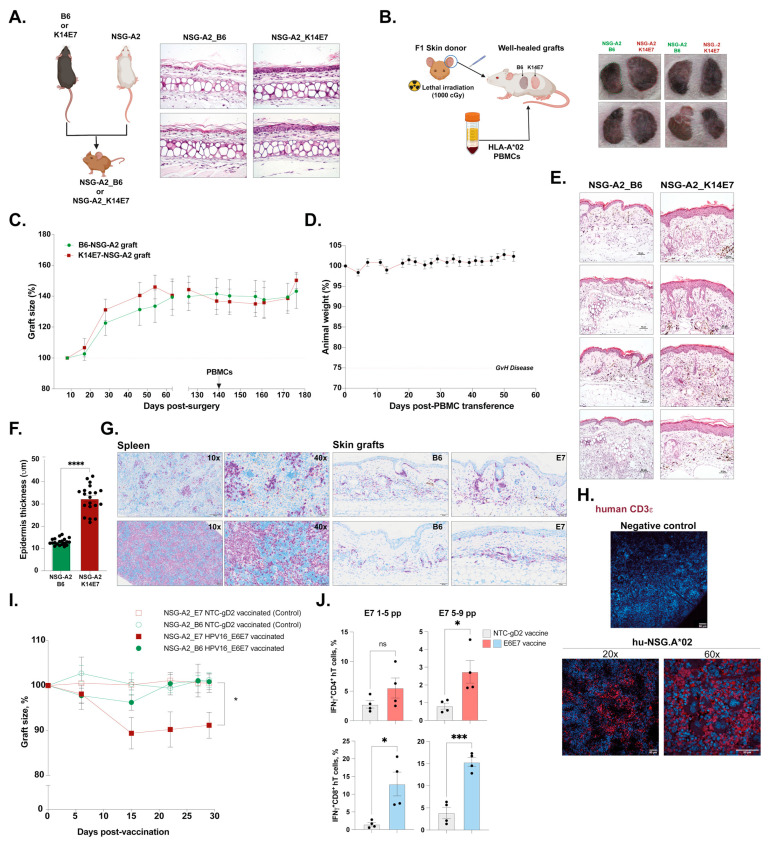
**Generation and testing of HPV-derived dysplastic skin-grafting humanized mouse model.** (**A**) Generation of skin expressing HPV16 E7 oncogene (NSG-A2_K14E7) and control (NSG.A2_B6) in the NSG-A2 background by crossing B6 or K14E7 animals with NSG-A2 mice (*left*). H&E staining of the ear skin of the F1 resulting from the cross B6 or K14E7 × NSG-A2 (*right*). (**B**) Schematic of the ear skin grafting process onto NSG-A2 recipients and transference of HLA.A*02+ matching PBMCs (*left*). Examples of controls (NSG-A2) and E7-expressing (NSG-A2_K14E7) well-healed grafts onto the flank of recipient mice (*right*). (**C**) Changes in graft size before and after PBMC transference up to 180 days post-skin-grafting. (**D**) Animal weight is expressed as the percentage of the initial weight. Signs of *GvHD* are typically manifested when 25% body weight loss is reached. (**E**) H&E staining of the skin grafts 2 months post-surgery. (**F**) Quantification of the thickness of the epidermis of skin grafts using H&E images and ImageJ software. The data are presented as mean values ± SEM. *p* = 0.1234 (ns), 0.0332 (*), 0.0021 (***), <0.0002 (****) (unpaired *t* test). (**G**) Immunohistochemistry showing human CD45^+^ infiltrating cells (purple) in the spleen and skin of grafted PBMC-hu-NSG-A2 animals. (**H**) Immunofluorescence staining showing human CD3e^+^ cells (red) infiltrating the spleen of skin-grafted control (no PBMCs were transferred) and humanized animals (PBMC-hu-NSG-A2). (**I**) Changes in graft size in control animals (NTC-gD2 vaccinated) and test animals (HPV16_E6E7 vaccinated). The data are presented as mean values ± SEM. *p* = 0.1234 (ns), *p* < 0.05 (*) (one-way ANOVA). (**J**) Percentage of IFNγ-expressing CD4^+^ and CD8^+^ human T cells (hT cells) resident in the spleen after *ex vivo* re-stimulation with two pools of five peptides each (Table 3, 1–5 and 5–9 pp), encompassing the entire length of the HPV16 E7 protein. The data are presented as mean values ± SEM. *p* < 0.05 (*), *p* < 0.001 (***), *p* < 0.0001 (****) (unpaired *t* test). pp = peptide.

**Figure 3 cells-12-02094-f003:**
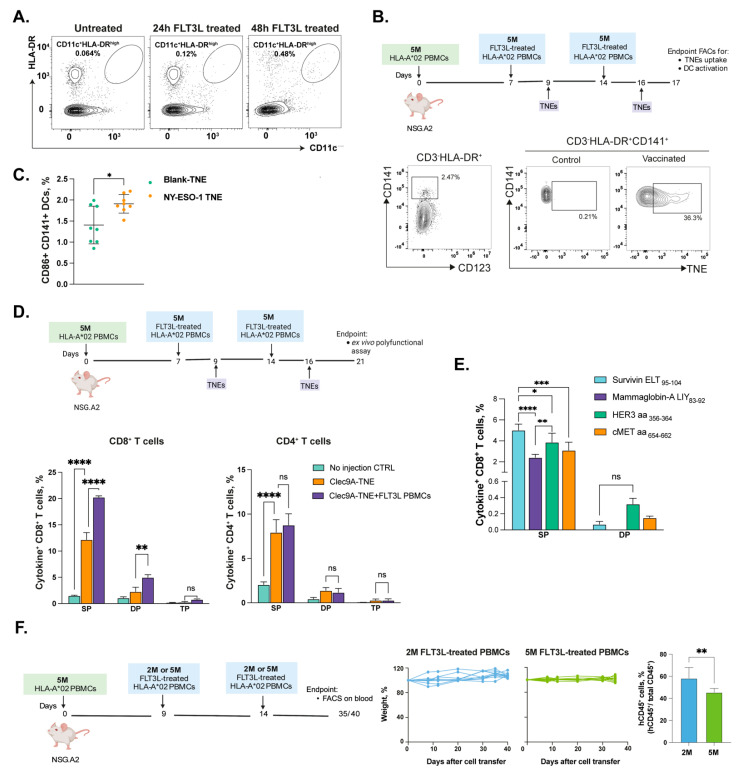
**Boosting of PBMC-hu-NSG-A2 mice with FLT3-L-treated autologous PBMCs.** (**A**) Representative dot plots of CD11c^+^HLA-DR^hi^ DCs expansion in human PBMCs after treating with 200 ng/mL of FLT3-L for 24 h or 48 h *in vitro*. (**B**,**C**) NSG-A2 mice were humanized with 5 × 10^6^ HLA-A*02^+^ PBMCs on day 0 and boosted with FLT3-L-treated autologous PBMCs on days 7 and 14. Dil-labelled F-actin-TNEs or F-actin-TNEs loaded with NY-ESO-1_119–143_ and NY-ESO-1_157–165_ epitope peptides were i.v. injected into the humanized mice on days 9 and 16. The spleens were collected on day 17 and processed as detailed in Methods. (**B**) Representative dot plots of F-actin-TNEs uptaken by splenic CD3^neg^HLA-DR^+^CD141^+^ DCs. (**C**) CD86 expression on CD141^+^ DCs one day after vaccination. (**D**) NSG-A2 mice were humanized with 5 × 10^6^ HLA-A*02^+^ PBMCs on day 0, with or without FLT3-L-treated autologous PBMCs boosting on days 7 and 14. F-actin-TNEs loaded with NY-ESO-1_119–143_ and NY-ESO-1_157–165_ epitope peptides were i.v. injected into the humanized mice on days 9 and 16. On day 21, the spleens were harvested and processed as described in Methods. Cytokine production in CD8^+^ and CD4^+^ T cells was measured by flow cytometry and intracellular cytokine staining after ex vivo stimulation of splenocytes with NY-ESO-1 epitopes for 6 h. (**E**) Experiment scheme as per (**D**) with FLT-L-treated autologous PBMCs boosting on days 7 and 14. F-actin-TNEs loaded with a pool of epitopes consisting of Survivin_95–104_, Mammaglobin_83–92_, HER3_356–364_, and cMET_654–662_ were i.v. injected into the mice on days 9 and 16. On day 21, the spleens were harvested and processed as described in Methods. Cytokine production in CD8^+^ and CD4^+^ T cells was measured by flow cytometry and intracellular cytokine staining after ex vivo stimulation of splenocytes with the individual epitopes listed. SP = single positive = cells expressing one of the cytokines IFNg, IL2, or TNFa; double positive = cells expressing any combination of two cytokines; TP = triple positive = cells co-expressing all three cytokines. (**F**) NSG-A2 mice were humanized with 5 × 10^6^ HLA-A*02^+^ PBMCs on day 0 and boosted with 2 × 10^6^ or 5 × 10^6^ FLT3-L-treated autologous PBMCs on days 7 and 14. Graphs show changes in weight and human CD45^+^ cell engraftment in mice that received different doses of FLT3-L-treated autologous PBMCs boosts. The data are presented as mean values ± SEM. *p* < 0.05 (*), *p* < 0.01 (**), *p* < 0.001 (***), *p* < 0.0001 (****) (two-way ANOVA).

**Figure 4 cells-12-02094-f004:**
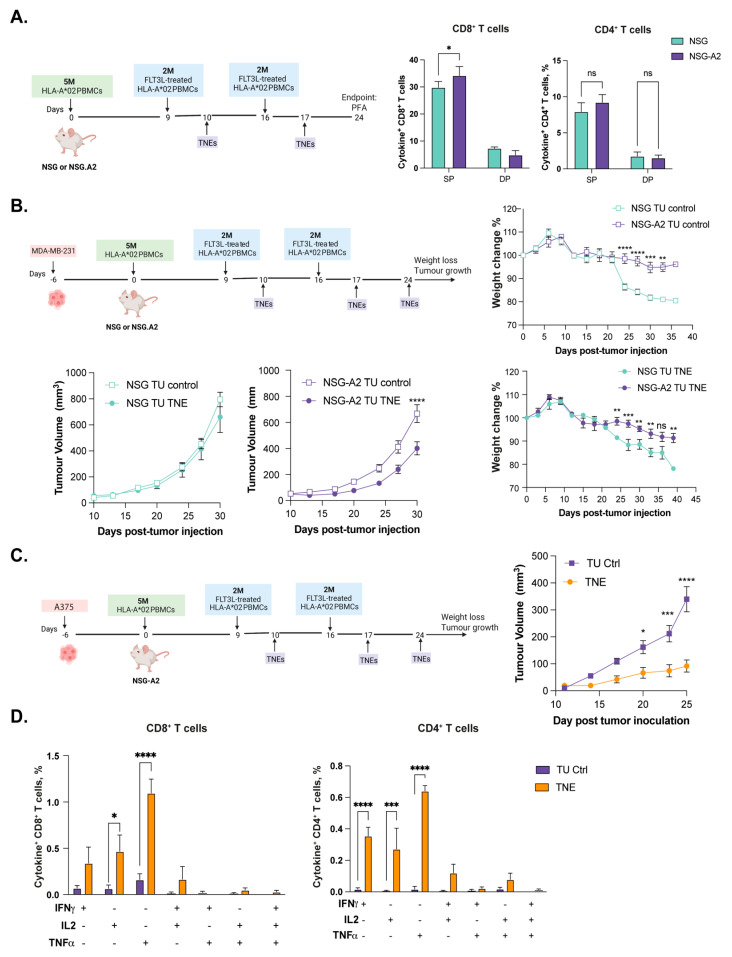
**The boosted PBMC-hu-NSG-A2 model supports therapeutic testing of cancer vaccines.** (**A**) NSG-A2 mice were humanized with 5 × 10^6^ HLA-A*02^+^ PBMCs on day 0 and boosted 2 × 10^6^ FLT3-L-treated autologous PBMCs on days 7 and 14. F-actin-TNEs loaded with hTERT_540–548_, Survivin_95–104_, and Survivin_97–111_ were i.v. injected into the humanized mice on days 10 and 17. On day 24, the spleens were harvested and processed as described in Methods. Cytokine production in CD8^+^ and CD4^+^ T cells was measured by intracellular cytokine staining and flow cytometry after ex vivo peptide stimulation. SP = single positive = cells expressing one of the cytokines IFNg, IL2, or TNFa; double positive = cells expressing any combination of two cytokines; TP = triple positive = cells co-expressing all three cytokines. (**B**) NSG or NSG-A2 mice were injected with MDA-MB-231 tumor cells orthotopically and humanized with HLA-A*02 PBMCs 6 days after (d0). All mice were boosted with FLT-L-treated autologous PBMCs on days 9 and 16. The mice were then vaccinated with F-actin TNEs loaded with the hTERT_540–548_, hSurvivin_95–104_, and hSurvivin_97–111_ epitopes on days 10, 17, and 24. Tumor growth and weight loss comparison of control (TU control) and vaccinated (TNE) NSG or NSG-A2 mice are reported. (**C**) NSG-A2 mice were injected with A375 human melanoma cells subcutaneously and humanized with HLA-A*02 PBMCs 6 days after (d0). The mice were boosted with FLT-L-treated autologous PBMCs on days 9 and 16. The mice were then vaccinated with F-actin TNEs loaded with a pool of tumor-associated immunogenic epitopes on days 10, 17, and 24. Tumor growth comparison in mice vaccinated with F-actin TNE (TNE) or unvaccinated (TU Ctrl) is reported. (**D**) The spleens were collected on day 40, and the cytokine production in splenic CD8^+^ and CD4^+^ T cells was measured by flow cytometry upon ex vivo stimulation with a pool of tumor-associated epitopes for 6 h and intracellular cytokine staining. The data are presented as mean values ± SEM *p* < 0.05 (*), *p* < 0.01 (**), *p* < 0.001 (***), *p* < 0.0001 (****) (two-way ANOVA). TU = tumor-bearing mice.

**Table 1 cells-12-02094-t001:** Tumor epitopes.

Tumor Model	Antigen	Type of Epitope	Aminoacidic Sequence
MDA-MB-231 TNBC	Survivin (97–111)	Promiscuous CD4 epitope	TLGEFLKLDRERAKN
	Survivin (ELT 95–104)	CD8 epitope	ELTLGEFLKL
	Telomerase (hTERT 540–548)	CD8 epitope	SLLDRFLATV
A375 melanoma	NY-ESO-1 (119–143)	Pan-DR epitope	PGVLLKEFTVSGNILTIRLTAADHR
	Tyrosinase (386–406)	Promiscuous CD4 epitope	FLLHHAFVDSIFEQWLQRHRP
	Cyclin-I	CD8 epitope	SLLDRFLATV
	Tyrosinase (369–377)	CD8 epitope	YMDGTMSQV
	STAT1	CD8 epitope	VLWDRTFSL
	Survivin (ELT 95–104)	CD8 epitope	ELTLGEFLKL
	MAGE-A3 (271–279)	CD8 epitope	FLWGPRALV

**Table 2 cells-12-02094-t002:** Antibodies for flow cytometry.

Antibody	Fluorochrome	Clone	Cat #	Company	Dilution
LIVE/DEAD™	Aqua	NA	L34957	ThermoFisher	1:1000
a-mouse-CD45.2	PE/Dazzle™	104	109846	BioLegend	1:200
a-human-CD45	Brilliant Violet 421™	2D1	368522	BioLegend	1:200
a-human-CD4	Alexa Fluor^®^ 700	RPA-T4	300526	BioLegend	1:200
a-human-CD8a	FITC	SK1	344704	BioLegend	1:200
a-human-CD19	APC	HIB19	302212	BioLegend	1:200
a-human-CD11c	Brilliant Violet 711™	3.9	301630	BioLegend	1:200
a-human-CD56	PE	QA17A16	985902	BioLegend	1:200
a-human-CD11b	PE/Cyanine7	LM2	393104	BioLegend	1:200
a-human-CD3e	APC	OKT3	317318	BioLegend	1:200
a-HLA.A2	APC	BB7.2	343308	BioLegend	1:200
a-human-TNFa	Alexa Fluor^®^ 647	MAb11	502916	BioLegend	1:100
a-human-IFNγ	Pacific Blue™	B27	506526	BioLegend	1:100
a-human-IL2	Alexa Fluor^®^ 700	MQ1-17H12	500320	BioLegend	1:100

**Table 3 cells-12-02094-t003:** HPV-16 E7 peptides (Mabtech AB).

Peptide	Sequence	Length (aa)
E7 #1 (1–22)	MHGDTPTLHEYMLDLQPETTDL	22
E7 #2 (11–32)	YMLDLQPETTDLYGYGQLNDSS	22
E7 #3 (21–42)	DLYCYEQLNDSSEEEDEIDGPA	22
E7 #4 (31–52)	SSEEEDEIDGPAGQAEPDRAHY	22
E7 #5 (41–62)	PAGQAEPDRAHYNIVTFCCKCD	22
E7 #6 (51–72)	HYNIVTFCCKCDSTLRLCVQST	22
E7 #7 (61–82)	CDSTLRLCVQSTHVDIRTLEDL	22
E7 #8 (71–92)	STHVDIRTLEDLLMGTLGIVCP	22
E7 #9 (77–98)	RTLEDLLMGTLGIVCPICSQKP	22

**Table 4 cells-12-02094-t004:** Antibodies used in the polyfunctional assay.

Antibody	Fluorochrome	Clone	Cat #	Company	Dilution
LIVE/DEAD™	Aqua	NA	L34957	ThermoFisher	1:1000
a-mouse-CD45.2	FITC	QA17A26	157608	Biolegend	1:200
a-human-CD45	PerCP	2D1	368506	Biolegend	1:200
a-human-CD3	Brilliant Violet 711™	OKT3	317328	Biolegend	1:200
a-human-CD4	PE/Cyanine7	RPA-T4	300512	Biolegend	1:200
a-human-CD8a	APC/Cyanine7	SK1	344714	Biolegend	1:200
a-human-TNFa	Alexa Fluor^®^ 647	MAb11	502916	Biolegend	1:100
a-human-IFNγ	Pacific Blue™	B27	506526	Biolegend	1:100
a-human-IL2	Alexa Fluor^®^ 700	MQ1-17H12	500320	Biolegend	1:100

## Data Availability

Files can be downloaded from UQ eSpace: https://rdm.uq.edu.au/files/6bfbd1a0-3b28-11ee-962e-fb3b15c008e8. https://doi.org/10.48610/988fb01.

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
