# Peer review of "Skin-Grafting and Dendritic Cell “Boosted” Humanized Mouse Models Allow the Pre-Clinical Evaluation of Therapeutic Cancer Vaccines"

_cells, 2023, doi:10.3390/cells12162094_

Round 1

Reviewer 1 Report

The manuscript presented for publication at Cells by Zeng et al. is providing novel humanized mice models for pre-clinical cancer vaccine research. More specifically, in their manuscript Zhen et al. showed that using fresh HLA matched PBMCs for reconstitution lead to a significantly increased engraftment of human immune cells, also in the skin graft model, without causing GvHD. In addition, vaccination was able to induce a reduction in the E7-expressing graft only, suggesting the induction of an antigen specific response. To increase the relevance for evaluation of cancer vaccine responses, the model was further expanded with the injection of Flt3L-expanded PBMCs to increase cDC1 presence. In addition, it was shown that HLA-matching was required to avoid the induction of GvHD. Using this model, the authors also showed an increased CD8 T-cell and anti-tumor response upon TNE treatment, showing the potential of this model for cancer vaccine research. Overall, this paper therefore provides the cancer vaccine research field with novel models to evaluate the efficacy of cancer vaccines and therefore is a great addition to the field. On top, the manuscript is complete and well written and therefore I would suggest accepting the manuscript upon revision of the minor comments indicated below.

Minor comments:

·        Line 320: The text refers to Figure 1D, while it should be 1E

·        In figure 3, the authors show the increase in DCs upon Flt3L treatment ex-vivo, however, it would be nicer if they could also show the increase in cDC1 specifically. Adding on to this, as follow-up, the authors should also show the increased presence of cDC1 in the skin grafts upon treatment with Flt3L treated PBMCs in comparison to when mice had extra doses of non-Flt3L treated PBMCs.

·        Figure 3D, E: The authors add in the legend the description of the abbreviations SP, DP and TP to clarify the figure.

·        Line 465: the authors indicate to the left panel to show GvHD, though this should be the right panel

Reviewer 2 Report

This study developed two modified humanized mouse models to evaluate the effectiveness of vaccines targeting Human Papillomavirus (HPV) antigens and delivering tumor antigens to human CD141+ Dendritic Cells (DCs). The models involved transferring HLA-A2+ human peripheral blood mononuclear cells (PBMCs) into severe immunocompromised NSG mice expressing HLA-A2 (Hu-PBL-NSG-A2), resulting in the engraftment of human adaptive immune cells over time without manifesting xeno-GvHD. Using these models, the authors created a hyperplastic skin graft model expressing the HPV16-E7 oncogene and observed rapid rejection upon vaccination with a DNA vaccine encoding HPV16-E6/E7 protein. Additionally, the study enhanced the abundance of CD141+ (Clec9A+) DCs by transferring additional PBMCs pre-treated with Flt3L, allowing to test the in vivo effect of Clec9A-targeting Tailored NanoEmulsions (TNE). The authors demonstrated potent antigen-specific T-cell responses and regression of triple-negative breast cancer and melanoma tumors following CD141+ DC-targeting vaccination. 

     Both humanized mouse models are unique and display in vivo effect of DNA vaccination. However, the mechanism to explain the impact of vaccination was not experimentally addressed well, particularly the induction of Ag-specific T cells by adaptively transferred DCs. Thus, some modifications are required to support the author's claim. 

Major comments

1.   The main concern of the submitted manuscript is that the authors did not address the benefit of the model the authors established in the study compared with other humanized mouse models, especially with a hematopoietic stem cell (HSC)-transplanted (Hu-SRC) NSG-A2 humanized mouse model (Najima et al., Blood 2016, PMID 26702062). Hu-SRC-NSG-A2 mice showed functional T cell differentiation and CD141+ DC differentiation. Therefore the model is available to test the induction of tumor antigen (WT1)-specific T cell responses in vivo. The authors should discuss the benefit and limitations of using PBMCs as donor cells in detail.

2.   Another concern is that adaptively transferred Flt3L–stimulated PBMCs. In this study, the authors injected Flt3L-stimulated PBMCs that contain an increased amount of CD141+ DCs compared with unstimulated PBMCs. Adaptively transferred PBMCs may repopulate in Hu-PB-NSG-A2 mice and respond to Clec9A-targeting Tailored NanoEmulsions (TNE) containing DNA vaccine for TAAs. The authors demonstrated that TNE vaccination induced anti-tumor responses and promoted cytokine production from T cells in Hu-PB-NSG-A2 mice. The consequence is interesting. However, the proof of concept of the strategy was not experimentally addressed; thereby, some questions remained. 1) Were FL-stimulated CD141+ cDC1s capable of inducing the proliferation of alloreactive T cells by MLR assay? 2) Did Ag-specific T cells induce in response to TNE-containing DNA vaccine? 3) Did transplanted CD141+ DCs indeed preferentially repopulate in the host lymphoid tissue as well as in the tumor? The authors should address them to demonstrate the advantage of the method. in addition, "boosted" is an overstatement considering the current result to describe in the title.

3.   Figure 1A: Preserved vs. fresh PBMCs–some essential information seems missing before making a conclusion. 

i) method for cryopreservation of PBMCs

ii) viability of thawed PBMCs

iii) frequency of each population in thawed PBMCs before injection

iv) cell yield and recovery rate after thawing

v) number of viable cells transplanted into NSG-A2 recipients

vi) irradiation dose

  All the above parameters could affect the consequence of PBMC engraftment.

4.   HPV16_E6E7 vaccination model (Fig. 2I): The cause of the reduction in size of skin graft needs to be addressed, particularly if HPV E6 or E7 Ag-specific T cells are implicated in the decrease in size. HLA-A2 tetramer assay would be the best to evaluate the effect of vaccination. If impossible, histological analysis for quantifying infiltrated T cells in the graft would also be available.

Minor comments

5.   Abstract: the abstract should be a total of 200 words maximum.

6.   What is the origin of DN T cells? A previous paper by Shultz et al. (PNAS 2010; PMID 20615947) demonstrated that they are γδ T cells in Hu-SRC-NSG-A2 mice.

7.   Figure 1(D); It isn't easy to recognize which clusters the authors mentioned in the main text. I would suggest numbering each cluster in addition to coloring it and describing the cluster number indicated in the text accordingly (e.g., lines 286-7, lines 289-90, and 293-294). 

8.   Improper use of terminology

i)    Fig. 1D; t-SNE would be correct.

ii)  Fig. 2. B6 is better to use instead of C57.

iii)Line 320; Figure 1D should be Figure 1E.

iv) Page 4, line 152; DNAse I change to DNase I.

v)  Page 13, line 449; PBMS-hu-NSG-A2 change to PBMC-hu-NSG-A2.

vi) Page 14, line 465; Figure 4B (left panel) change to (right panel).

vii) Figure S1 B. Left panel, an extra word (F).

Round 2

Reviewer 2 Report

Most of our concerns have been addressed in the current version of the manuscript except for one minor point regarding the irradiation method.

Minor comment

Irradiation method: This reviewer would like to make sure if NSG-A2 mice were irradiated upon PBMC injection. If they were not irradiated for establishing Hu-PBL-NSG-A2 mice, the authors would indicate “non-irradiated” NSG-A2 mice in line 124 (Page 3). In addition, the lethal irradiation dose must be 1000 cGy (=10 Gy). The authors should correct the dose.

Author Response

We would like to thank Reviewer 2 for the very on-point corrections. We have taken them into account and improved the manuscript accordingly (changes in red).

Minor comment

Irradiation method: This reviewer would like to make sure if NSG-A2 mice were irradiated upon PBMC injection. If they were not irradiated for establishing Hu-PBL-NSG-A2 mice, the authors would indicate “non-irradiated” NSG-A2 mice in line 124 (Page 3). In addition, the lethal irradiation dose must be 1000 cGy (=10 Gy). The authors should correct the dose.

Response: we would like to confirmed that the recipient animals were not irradiated, only the skin donors. To clarify this point we have changed the text on page 3, line 124: [Afterwards, PBMCs were immediately injected into non-irradiated NSG-A2 mice for engraftment or cryo-preserved] and  line 131: [Briefly, whole ears from lethally irradiated (1000 cGy) donor mice (NSG-A2_C57BL/6J and NSG-A2_K14E7) were surgically removed and split into dorsal and ventral surfaces. The remaining cartilage tissue was gently removed with a scalpel, and two dorsal sheets were placed onto an ~1cm2 incision on each thoracic flank of fully anesthetized non-irradiated NSG-A2 recipients].